# Investigation of Carbon Fiber on the Tensile Property of FDM-Produced PLA Specimen

**DOI:** 10.3390/polym14235230

**Published:** 2022-12-01

**Authors:** Mengyu Cao, Tianqi Cui, Yuhang Yue, Chaoyu Li, Xue Guo, Xin Jia, Baojin Wang

**Affiliations:** 1College of Mechanical Science and Engineering, Northeast Petroleum University, Daqing 163318, China; 2Procurement & Equipment Department, China National Petroleum Corporation, Beijing 100007, China

**Keywords:** tensile property, fused deposition modeling, carbon fiber, polylactic acid

## Abstract

Herein, the effect of carbon fibers (CFs) on the tensile property of a polylactic acid (PLA) specimen prepared by utilizing the fused deposition modeling (FDM) method, is investigated. The tensile property, crystal structure, and morphology of FDM-produced specimens were detected by universal testing machine, X-ray diffraction (XRD), and scanning electron microscopy (SEM), respectively. Meanwhile, the reinforcement mechanism of CFs on the FDM-printed PLA specimens was also studied. The DSC curves indicated that the crystalline structure of the PLA-CF specimen was higher than the PLA specimen. After the introduction of CFs, the XRD results showed the crystal structure of PLA varied from non-crystalline to α crystalline, and the SEM results illustrated the terrible bonding interface between carbon fiber and PLA. Interestingly, after the introduction of carbon fiber, the tensile strength of the PLA specimen reduced from 54.51 to 49.41 MPa. However, compared with the PLA component, the Young’s modulus and the elongation-at-break of the PLA-CF specimen increased from 1.04 GPa and 6.26%, to 1.26 GPa and 7.81%, respectively.

## 1. Introduction

Fused deposition modeling (FDM) technology is also known as fused filament fabrication (FFF) or fused layer modeling (FLM). As one of the current advanced manufacturing technologies, this method has been widely used in various commodity manufacturing processes [1,2,3]. This FDM method can achieve a layered stacking of materials. With this method, the digital model can be quickly printed into a high-quality part [4]. Compared with traditional manufacturing methods, FDM has many advantages. For example, the FDM method has higher automation, lower cost and more convenient operation [5]. Therefore, the FDM method has been widely used in the manufacturing of automobile parts, aviation materials and structures, food packaging and medical devices [6]. Since the FDM method uses a layer-by-layer accumulation method to prepare samples, thermoplastic polymer filament is very suitable as its raw material [7]. Polylactic acid (PLA), acrylonitrile butadiene styrene (ABS) and polycarbonate (PC) are commonly used thermoplastic polymer filaments. Among these polymer materials, polylactic acid (PLA) has higher safety, stronger degradability and easier access to raw materials. Therefore, its application is relatively more extensive [8]. Polylactic acid (PLA) is essentially an engineering plastic. However, compared with other engineering plastics, the toughness of polylactic acid (PLA) is relatively poor. This disadvantage also limits the application scope of polylactic acid (PLA), to a certain extent.

Many methods (i.e., optimization of process parameters, grafting, and blending modification) have been used to improve the mechanical property of the FDM-produced PLA specimen [9]. For example, some PLA parts were prepared by Behzadnasab et al. [10] though the FDM method, and the influence of nozzle temperature and filling method on the mechanical properties of parts, were studied. Hu et al. [11] designed a mathematical model for describing the operation parameters of the mechanical properties of FDM-PLA with chopped long carbon fibers. Li et al. [12] established a mathematical model to reveal the strengthening mechanism of continuous carbon fibers on FDM-PLA samples. Maqsood et al. [13] studied the tensile and flexural properties of PLA carbon fiber, PLA-short carbon fiber (SCF), PLA-continuous carbon fiber (CCF) and PLA-SCF-CCF, during the FDM process. Reverte et al. [14] reported the influence of short carbon fiber on the mechanical performance and ILSS of a PLA sample fabricated by FFF technique. In these studies, the blending modification of CFs in PLA has been widely used due to its convenient operation and good effect. Hence, CFs were also used for enhancing the tensile property of the PLA component in this paper. However, research papers which focused on the tensile property of FDM-PLA components using short CFs, were few. In addition, the reinforcement mechanism of CFs on the FDM-printed PLA components has not been further discussed.

In this article, the PLA and PLA-CF component were both fabricated by FDM technique. The tensile properties of the two types of specimens were measured by a universal testing machine. The influence of short CFs on the structure and crystallization of the PLA component was detected by X-ray diffraction and differential scanning calorimeter, respectively. The fracture morphology of specimens after the tensile test, and the cross section morphology of filaments, were measured by scanning electron microscope. The reinforcement mechanism of the carbon fiber on the PLA specimen was also analyzed.

## 2. Materials and Methods

### 2.1. Preparation

In this work, the PLA pellets and CF were purchased from Caige Consumables Co. Ltd. (Nantong, China) and Zhiweida Co., Ltd. (Suzhou, China), respectively. The size of carbon fiber was 30–50 μm and the diameter was 10–15 μm. The carbon fiber was not further disposed by chemical agent. The PLA pellets were melted and blended with carbon fiber through a TY-7004 single screw extruder (purchased from Tianyuan Co., Ltd., Suzhou, China) under the condition of 200 °C and 30 r/min. The PLA-CF filament had 30 V% carbon fiber and the remainder was PLA. The diameters of the PLA and PLA-CF filaments were both 1.75 mm. Cross sectional images of the PLA and PLA-CF filaments are shown in Figure 1. It was observed that the carbon fibers were distributed in the PLA-CF filament, and the agglomeration of carbon fiber emerged in the cross section of the PLA-CF filament. Meanwhile, no obvious tiny pores appeared in the PLA and PLA filament cross-section. A DF-G3545 FDM printer (purchased from Zhuhai Dufen Automation Technology Co., Ltd., Zhuhai, China) was applied to fabricate the PLA and PLA-CF specimens, as shown in Figure 2. The schematic process of the FDM printer is displayed in Figure 2a. The FDM printer was primarily composed of filament feed-stock, extrusion head, servo device, and working platform (Figure 2b). During the FDM process, the filament feed-stock was delivered into the extrusion head by a pair of feed rolls. The filament feed-stock was heated to semi-molten pattern and extruded from the nozzle at the pressure of gravity. Based on the design of G-codes, the semi-molten filament feed-stock moved along a specific route and was deposited on the working platform to form the layer-stacked specimen [15]. According to the previous literature, the main preparation parameters for the specimen are listed in Table 1.

### 2.2. Characterization

A differential scanning calorimeter (DSC, NETZSCH 204F1, purchased from BiBiH International Trading (Shanghai) Co., Ltd., Shanghai, China) was used to measure the crystallization and melting behavior of FDM-produced samples. The sample, in an atmosphere of N_2_ gas, was firstly heated from room temperature to 220 °C, at a rate of 20 °C/min. The sample was kept at 220 °C for 3 min, with the objective of eliminating thermal history. Then, the sample was cooled to 40 °C at a rate of 10 °C/min. Finally, the DSC curves of the sample were recorded and analyzed. The crystallinity ($\chi_{c}$) of the PLA and PLA-CF samples were calculated using the following equation:(1)$$\chi_{c}=\frac{\Delta H}{\Delta H_{100}}$$
where ΔH was the enthalpy of tested crystallization and $\Delta H_{100}$ was the standard enthalpy of full crystallization, which is 95.0 J/g for PLA.

The fracture surfaces of the PLA and PLA-CF specimens were examined by a Quanta FEG450 scanning electron microscope (purchased from Beijing Yuanhaiwei Technology Co., Ltd., Beijing, China). DX-100 X-ray diffraction equipment (purchased from Wuhan Lanruida Information Technology Co., Ltd., Wuhan, China) was used to detect the crystal structure of the specimens. The working condition of the XRD facility was set at Cu Kα (λ = 0.154 nm), 2 kV, and the scanning scope (2θ) from 1° to 40° at a rate of 2°/min.

Figure 3 shows the sizes and the raster angle of specimens in detail, which accorded to the standard of ASTM D638-14. The schematic and experimental pictures of the universal testing machine are displayed in Figure 4. A CMT 5105 universal testing machine (purchased from Changchun Haoyuan Testing Machine Co., Ltd., Changchun, China) was used to measure the tensile properties of the PLA and PLA-CF specimens. Each specimen was measured 5 times during tensile testing, and the final results were averaged. The cross-head speed was 3 mm/min under the condition of static load and indoor temperature. Meanwhile, the elongation-at-break of specimens was calculated by the migration length of the cross-head.

## 3. Results and Discussion

### 3.1. DSC Curves Analysis

Figure 5 displays the DSC curves of the PLA and PLA-CF specimens. There was no obvious cool crystallization peak in the PLA specimen. Three obvious peaks appeared in the DSC curve of the PLA-CF specimen in the following order: glass transition peak, cool crystallization peak, and melt peak. The cool crystallization peak emerged in the PLA-CF specimen, rather than in the PLA, which illustrated that the crystallization ability and degree of pure PLA were relatively weak.

It can be seen from Table 2 that the glass transition temperature (Tg) of PLA-CF (60.23 °C) was slightly higher than for PLA (59.46 °C). Although the carbon fiber and PLA were independent, the CFs in the PLA-CF could hinder the movement of the PLA chain as the temperature increased, which caused the thermal stability of PLA-CF to improve. The melt temperature (Tm) of PLA was lower, whereas the cool crystallization temperature (Tc) of PLA-CF was higher than pure PLA, which meant that the various orientations of carbon fiber dispersed in the blends may produce a heterogeneous nucleation effect on the crystallization of PLA. These results were similar to those of Liu et al. [16] regarding the improvement of carbon fiber on the thermal stability and crystallization of PLA material.

### 3.2. XRD Detection Results

Figure 6 indicates XRD patterns of PLA and PLA-CF specimens. The diffraction peak of the PLA specimen appeared at 18.3°, which meant that the PLA specimen was non-crystalline [17]. The diffraction peaks of the PLA-CF specimen appeared at 16.7° and 19.0°, respectively, corresponding to (200) and (203) crystal planes, and, therefore, was typical α crystal [17]. Compared with PLA, an obvious variation emerged in the position and intensity of the PLA-CF diffraction peak, which indicated that the crystal structure of PLA was changed after the incorporation of CFs. Meanwhile, the diffraction peak of PLA-CF at 16.6° and 19.0° widened, which showed that the CFs refined the crystal size of the PLA [18]. The result was attributed to the introduction of CFs which increased the crystallinity of PLA via a process of heterogenous nucleation. In addition, the diffraction peak of carbon fiber was not found in the results shown in Figure 6. This was because the diffraction peak intensity of the carbon fibers was so low, that it was covered by the diffraction peak intensity of the PLA.

### 3.3. Tensile Performance Measurement

Figure 7 presents the stress–strain curves of PLA and PLA-CF specimens in tensile testing. The stress–strain curve of the PLA specimen grew almost linearly with strain, and fractured at its peak. By contrast, the stress–strain curve of the PLA-CF specimen differed substantially from the PLA specimen. The PLA-CF stress–strain curve could be divided into two stages of yield point and stress platform. During the tensile test, the stress–strain curve of the PLA-CF specimen exhibited linearly at primary stage. After the stress grew to the peak, it held steady as the strain increased. Analyzing the results showed that the fracture behavior of the PLA-CF specimen varied from brittle fracture to ductile fracture, after the carbon fiber was incorporated into the PLA substance.

Figure 8 shows the tensile properties of PLA and PLA-CF specimens, and their detailed data are listed in Table 3. As shown in Figure 8 and Table 3, the tensile properties of the FDM-produced specimen were obviously influenced by carbon fiber. The maximum tensile strength, Young’s modulus, and elongation-at-break of the PLA specimen were 54.51 MPa, 1.04 GPa, and 6.26%, respectively. After carbon fiber was introduced into the PLA substrate, the tensile strength, Young’s modulus, and elongation-at-break of the PLA-CF specimen increased to 49.41 MPa, 1.26 GPa, and 7.81%, respectively. The results can be explained as follows. Firstly, the different orientations and the terrible bonding interface of CFs dispersed in PLA decreased the continuity of the PLA-CF material, which led to the tensile property of PLA-CF sample being weaker than that of the PLA sample. This result is consistent with Figure 9. Meanwhile, the agglomeration of carbon fiber would decrease the tensile strength of the PLA-CF specimen. The proposal by Haafiz et al. [19], that the agglomeration of cellulose nanowhiskers (CNW) decreased the tensile strength of PLA/CNW bionanocomposites, has been proved by this conclusion. Secondly, the Young’s modulus of carbon fiber was much stronger than that of PLA, which resulted in the Young’s modulus of the PLA-CF specimen being higher than the PLA specimen. Thirdly, during the printing process of the PLA-CF sample, the existence of carbon fiber aggravated the crystalline transition and volume shrinkage of the PLA, which caused the internal shrinkage stress to constantly be accumulated [20]. These reasons resulted in the tensile strength of PLA-CF specimen being lower than that of PLA, whereas its Young’s modulus and elongation-at-break were—on the contrary—high.

### 3.4. Fractured Morphology Observation

Figure 9 illustrates the SEM images of the fractured surface of PLA and PLA-CF specimens. The fractured surface of the PLA specimen exhibited as smooth and flat, as shown as Figure 9a. However, Figure 9b displays the various orientations of CFs dispersed in the PLA substrate; obvious tiny pores and some broken carbon fibers emerged in the fractured surface of the PLA-CF specimen. The result was attributed to the carbon fibers being pulled out of the PLA substrate under the implementation of an external force, and the tiny pores were generated on the fracture surface of PLA-CF specimen [21]. In addition, the gaps between the carbon fibers and PLA substrate were distinct, indicating the bonding interface was terrible.

### 3.5. Reinforcement Mechanism

Figure 10 reveals the reinforcement mechanism of carbon fiber on the PLA substrate. Under the quasi-static load and room temperature, both sides of the FDM-produced specimens were continuously stretched by force until they fractured. During this process, the carbon fiber acted as a crucial reinforcement for enhancing the toughness of the PLA specimen.

The reason for this could be explained as follows. Firstly, during the tensile test, the different orientations of carbon fiber in the PLA were able to absorb and transfer the external load, which resulted in the Young’s modulus and elongation-at-break of the PLA-CF specimen. This result correlated with the study by Li et al. [22] regarding carbon nanotubes, on the Young’s modulus and elongation-at-break of PLA. Secondly, the introduction of the carbon fiber acted as the function of heterogenous nucleation, which increased the crystallinity of the PLA and refined the grain size of the PLA around the carbon fiber. This conclusion was similar to the research of Jin et al. [23] which studied reinforcement by ramie fiber of the heterogenous nucleation of PLA. Finally, the crystallinity of the PLA-CF specimen was higher than the PLA specimen, which caused the Young’s modulus and elongation-at-break of the PLA-CF specimen to be fine. This result was proved by the study by Liao et al. [24] regarding how crystallinity growth was used to improve the mechanical properties of a PLA specimen.

## 4. Conclusions

(1)DSC curves showed that the crystallinity of the PLA-CF specimen was higher than the PLA sample. The SEM results indicated that the agglomeration of carbon fibers emerged in a cross-section of the PLA-CF filament, and the terrible bonding interface between carbon fiber and PLA;(2)XRD patterns illustrated the crystal structure of the PLA specimen varied from non-crystalline to α crystalline after the carbon fiber was introduced. Meanwhile, the diffraction peak of PLA-CF at 16.6° and 19.0° widened, which showed that CFs refined the crystal size of PLA;(3)After the incorporation of carbon fiber, the tensile strength of the PLA specimen reduced from 54.51 to 49.41 MPa, while the Young’s modulus and elongation-at-break increased from 1.04 GPa and 6.26%, to 1.26 GPa and 7.81%, respectively.

## Figures and Tables

**Figure 1 polymers-14-05230-f001:**
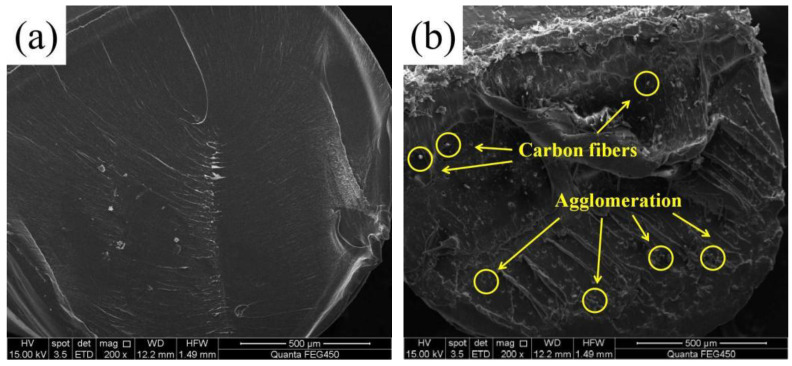
Cross-sectional SEM images of filament: (**a**) PLA and (**b**) PLA-CF.

**Figure 2 polymers-14-05230-f002:**
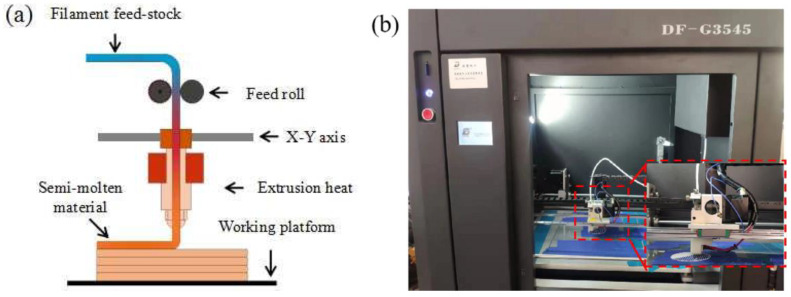
(**a**) Schematic FDM process, and (**b**) DF-G3545 FDM printer.

**Figure 3 polymers-14-05230-f003:**
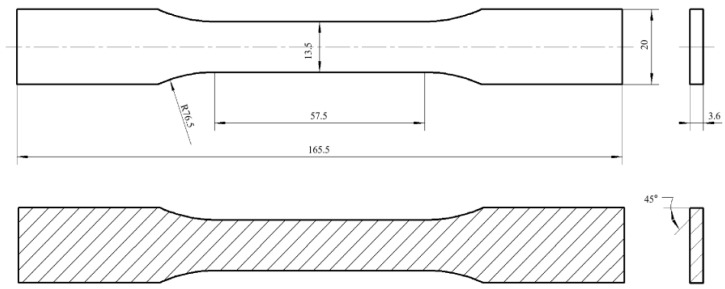
Size and raster angle of the FDM-produced specimen (unit, mm).

**Figure 4 polymers-14-05230-f004:**
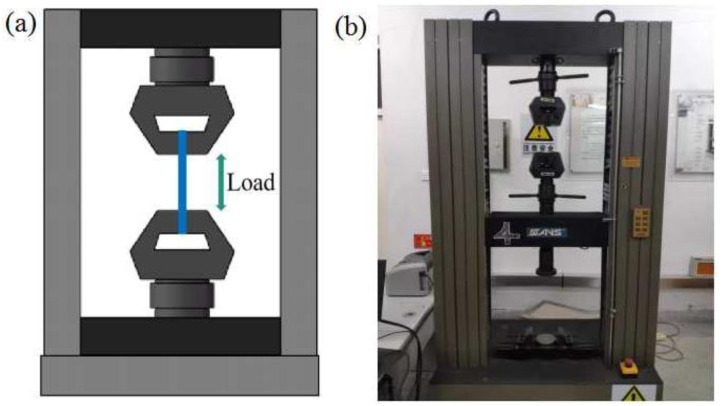
Tensile testing schematic and equipment: (**a**) schematic and (**b**) equipment.

**Figure 5 polymers-14-05230-f005:**
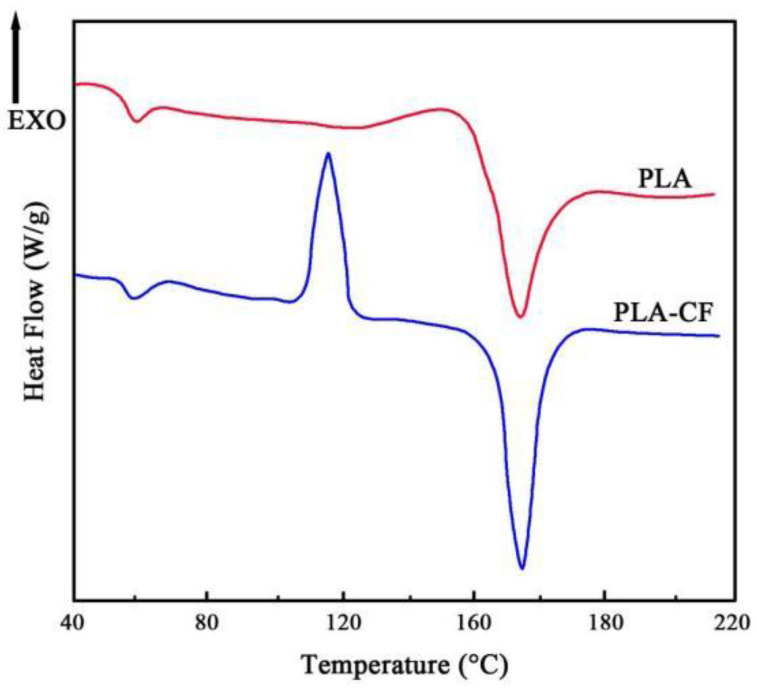
DSC curves of FDM-produced specimens.

**Figure 6 polymers-14-05230-f006:**
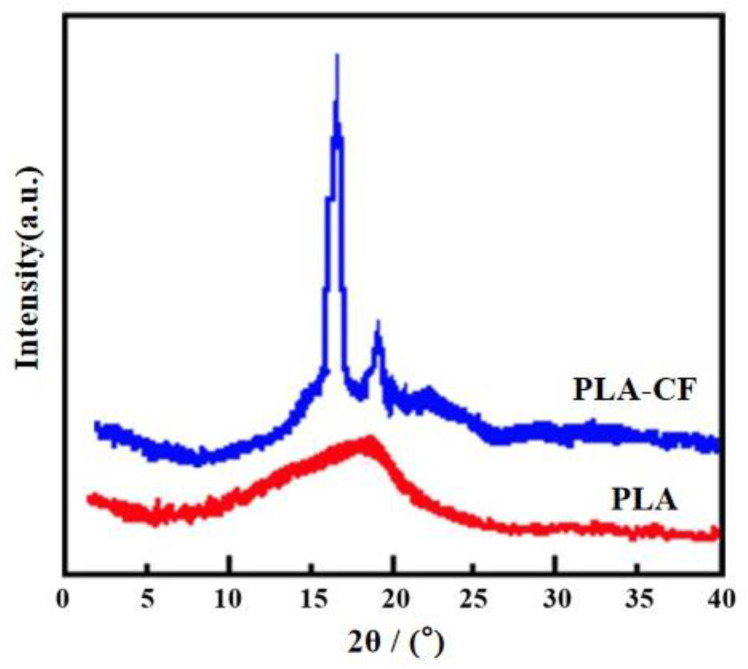
XRD patterns of specimens.

**Figure 7 polymers-14-05230-f007:**
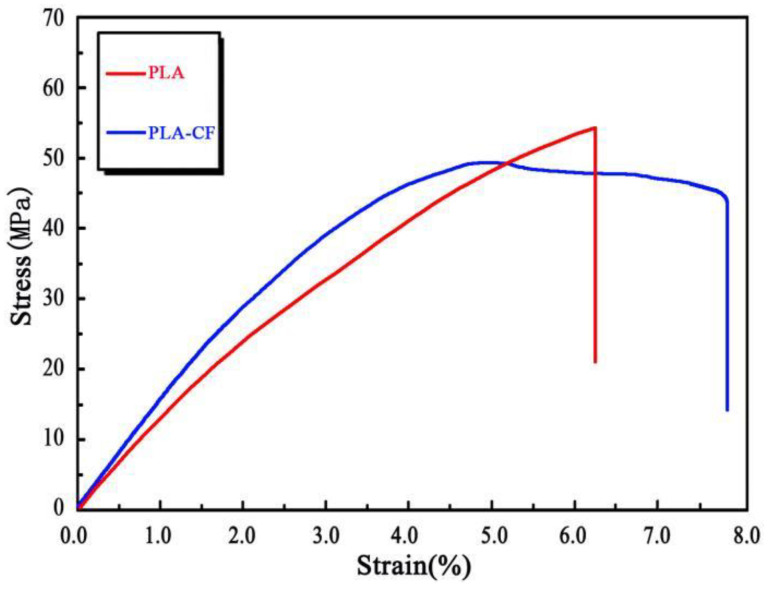
Tensile stress–strain curves of FDM-produced specimens.

**Figure 8 polymers-14-05230-f008:**
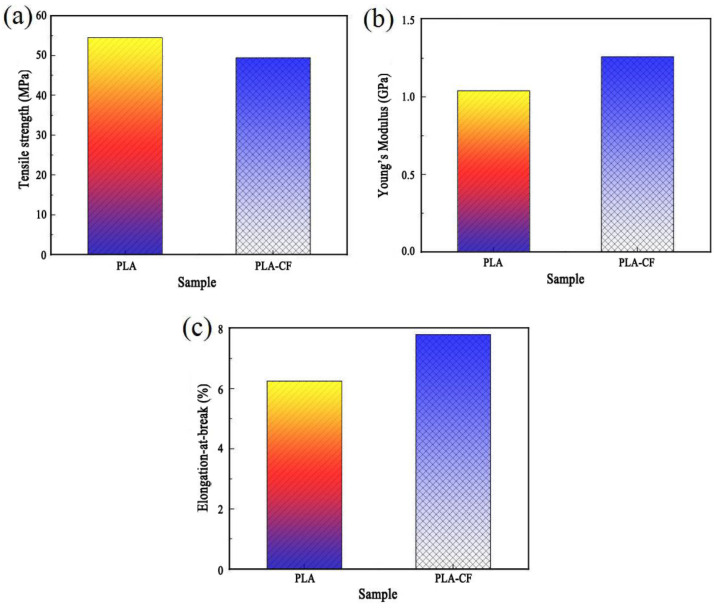
Tensile properties of FDM-produced samples: (**a**) tensile strength, (**b**) Young’s modulus, and (**c**) elongation-at-break.

**Figure 9 polymers-14-05230-f009:**
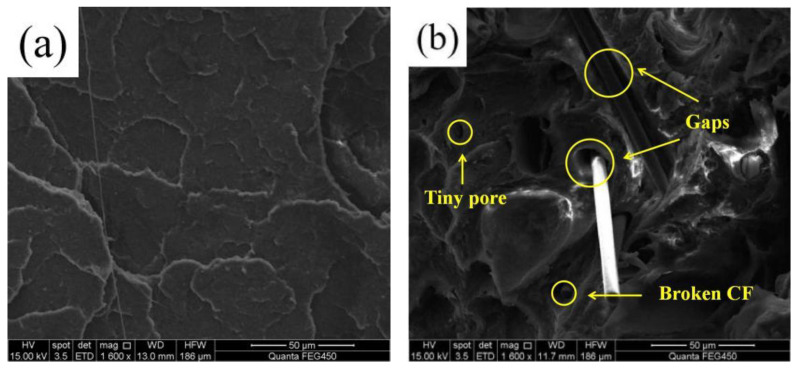
SEM images of fractured surface of (**a**) PLA, and (**b**) PLA-CF specimen.

**Figure 10 polymers-14-05230-f010:**
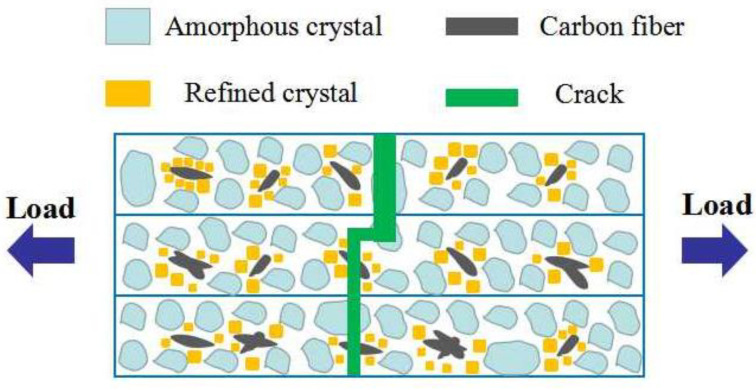
Reinforcement mechanism of carbon fiber in the PLA substrate.

**Table 1 polymers-14-05230-t001:** Main preparation parameters of the FDM-produced specimen.

Specimen	PLA	PLA-CF
Printing speed (mm/s)	5	5
Print temperature (°C)	210	210
Platform temperature (°C)	60	60
Nozzle diameter (mm)	0.4	0.4
Layer thickness (mm)	0.9	0.9
Feed-stock speed (mm/s)	10	10
Infill density (%)	100	100
Filament distance (mm)	0.4	0.4

**Table 2 polymers-14-05230-t002:** Thermal properties of PLA and PLA-CF specimens.

Specimens	Tg (°C)	Tc (°C)	Tm (°C)	*χ*_c_ (%)
PLA	59.46	133.05	164.91	4.15
PLA-CF	60.23	110.63	166.87	9.01

**Table 3 polymers-14-05230-t003:** Detailed data of FDM-produced specimens in tensile testing.

Specimen	PLA	PLA-CF
Tensile strength (MPa)	54.51	49.41
Young’s Modulus (GPa)	1.04	1.26
Elongation-at-break (%)	6.26	7.81

## Data Availability

Not applicable.

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
