# Peer review of "Investigation of Carbon Fiber on the Tensile Property of FDM-Produced PLA Specimen"

_polymers, 2022, doi:10.3390/polym14235230_

Round 1

Reviewer 1 Report

The article is a comparative study of the mechanical properties of PLA specimens and carbon fiber reinforced PLA specimens prepared via fused filament fabrication (FFF). Though there are some experimental findings reported, I did not find any novelty in this article. The paper has many formatting and grammatical errors in it. Here are some suggestions to increase the quality of the paper.  

  1. Please explain the novelty of the research clearly in the abstract and the major conclusions.
  2. Please use the terminology correctly. In AM, FFF is fused filament fabrication, not fused manufacturing.
  3. Please define everything for the first time before using the abbreviated form (e.g., lines 46 and 48).
  4. The problem and objectives of the research are not well defined.
  5. The material section lacks most of the details. For a short-fiber composite, it is essential to report the volume fraction or weight fraction of carbon fiber used, as well as the size and aspect ratio of the fiber. These details are missing in the material section.
  6. The purpose of figure 1 is not clear. A detailed explanation of the attributes seen in the SEM image needs to be included.
  7. The author has calculated the Young’s modulus that needs strain. In the material section, the author needs to clearly mention what equipment was used to measure the strain.
  8. The fiber is supposed to impart brittleness to the specimen. The opposite is achieved by the test result. Insight on the reason for this behavior is missing.
  9. The author needs to carefully read the article before submission. The detail in the results section does not correspond to the figure.
  10. The bar graph in Figure 8b shows that the Young’s modulus of the PLA specimen with carbon fiber is higher compared to a neat PLA specimen. However, in contrast, figure 7 shows a different result. The slope in figure 7 looks almost the same.

Reviewer 2 Report

The article "Investigation of carbon fiber on the tensile property of FDM- produced PLA specimen" presents experimental tensile results, XRD, and SEM of  CF-PLA composites. The layout/flow of the manuscript is proper but can be further improved. The authors had conducted reasonable number of tests. To further improve the article, authors are encouraged to consider the following points.

1) Line 23, FDM should be fused deposition modeling, instead of Fusion deposition modeling. FFF is refer for Fused Filament Fabrication.

2) Background/ Literature review can be improved. Currently refs (1-16) were mentioned during introduction and without detailed discusision. (!16 out of 30 references are for introduction!). Suggest to reduce the references for general introduction and include more relevent works. For example similar research works on CF-PLA reported previously should be discussed in more detail. Some samples works are 

https://doi.org/10.1016/j.jcomc.2021.100112 .  your cited it as ref 27 Maqsood et al [27]. This work is very related to yours and should be highlighted.

https://doi.org/10.3390/ma14227062

https://doi.org/10.1016/j.addma.2018.12.010

and many more.

3) Please define SCF CCF at line 46.

4) Grammatical errors were detected. example

line 60 - The fracture mechanism of specimen and the reinforcement mechanism of carbon fiber on the PLA specimen was also analyzed.

line 64- the PLA and PLA-CF filaments at a diameter of 1.75 mm was both

and more.

4) materials and methods

Some important settings during sample preparation (3D printing) were not reported. For example what are the printing infill pattern, infill density, wall thickness, layer thickness etc of the print?  

what is the % CF content in CF-PLA filament?

without this info, no one can duplicate your results.

5) Fig 1b, it will be great if you can label CF in the figure.

6) Fig 3; please provide unit.

7) line 180-' many tiny pores and some broken carbon fibers emerged in the fractured surface of PLA-CF specimen. When the carbon fibers were pulled out from PLA substrate under the implement of external force, these tiny pores generated on the fracture surface of PLA-CF specimen.'

Currently only SEM of fracture surfaces were provided. Can you confirm your argurment? Can you detect tiny hole in SEM images of CF-PLA printed part before fracture? Are the hole sizes similar to the diameter of CF?

8) Fig 10-  please label green, blue, yellow and borwn objects of your Fig 10. 

9) Please compare your results with previous works.

Reinforcement Mechanism- please provide more support for this mechanism.

10) Please confirm your conclusion

elongation-at-break and Young’s modulus of PLA-CF specimen increased to 1.26 GPa and 7.81%

Round 2

Reviewer 1 Report

The article looks much better now. 

Reviewer 2 Report

the authors have addressed all previous comments. Thank you.